# Introduction of Reversible Urethane Bonds Based on Vanillyl Alcohol for Efficient Self-Healing of Polyurethane Elastomers

**DOI:** 10.3390/molecules24122201

**Published:** 2019-06-12

**Authors:** Dae-Woo Lee, Han-Na Kim, Dai-Soo Lee

**Affiliations:** Division of Semiconductor and Chemical Engineering, Chonbuk National University, Baekjedaero 567, Jeonju 54896, Korea; dwlee2310@hanmail.net (D.-W.L.); hnk07@hanmail.net (H.-N.K.)

**Keywords:** vanillyl alcohol, self-healing efficiency, mechanical property, reversible urethane bond

## Abstract

Urethane groups formed by reacting phenolic hydroxyl groups with isocyanates are known to be reversible at high temperatures. To investigate the intrinsic self-healing of polyurethane via a reversible urethane group, we synthesized vanillyl alcohol (VA)-based polyurethanes. The phenolic hydroxyl group of vanillyl alcohol allows the introduction of a reversible urethane group into the polyurethane backbone. Particularly, we investigated the effects of varying the concentration of reversible urethane groups on the self-healing of the polyurethane, and we proposed a method that improved the mobility of the molecules contributing to the self-healing process. The concentration of reversible urethane groups in the polyurethanes was controlled by varying the vanillyl alcohol content. Increasing the concentration of the reversible urethane group worsened the self-healing property by increasing hydrogen bonding and microphase separation, which consequently decreased the molecular mobility. On the other hand, after formulating a modified chain extender (m-CE), hydrogen bonding and microphase separation decreased, and the mobility (and hence the self-healing efficiency) of the molecules improved. In VA40-10 (40% VA; 10% m-CE) heated to 140 °C, the self-healing efficiency reached 96.5% after 30 min, a 139% improvement over the control polyurethane elastomer (PU). We conclude that the self-healing and mechanical properties of polyurethanes might be tailored for applications by adjusting the vanillyl alcohol content and modifying the chain extender.

## 1. Introduction

Polyurethane elastomer (PU) is a versatile polymer with many industrial applications. PUs is generally synthesized from polyols, diisocyanate, and a chain extender by the prepolymer method. The physical properties of PU can be adjusted by varying the types and amounts of polyols, isocyanates, and chain extenders in the precursor mixture. PU is a segmented block copolymer composed of a soft domain (polyols) and a hard domain (isocyanate and the chain extender). The polyol in the soft domain gives the PU its elastic quality, while the hard domain (depending on its nature) confers hardness and rigidity. Thus, microphase separations and the contents of the soft and hard domains significantly affect the mechanical properties of PU. Thermal stability of PU has also been reported. Chain scission occurs in the hard segment when PU is heated above 200 °C [1,2,3]. PU is widely used as thermal insulation foams and as adhesives, fibers, coatings, shock-absorbers, and elastomers in industrial fields [4,5,6,7].

Recently, many studies have focused on the self-healing and shape memory properties of PU. Self-healing PUs are materials capable of repairing damage via release of healing agents from embedded microcapsules or a temporary increase in mobility leading to re-flow of the material in the damaged area. Compared to other materials, there are many choices to impart self-healing properties to Pus, as the physico-chemical properties of PU can be adjusted with the types and amounts of polyols, isocyanates, and chain extenders. Many researchers are improving the durability and applicability of polymer products by enhancing their self-healing properties. Various fabrication methods for self-healing polymers have been suggested. In early research, self-healing was extrinsically enhanced by dispersing microcapsules in the polymer [8]. When cracks grew in the polymer, microcapsules were broken and released a healing agent that filled the cracks and healed the polymer. However, owing to the low storage stability of the healing agent stored in the microcapsule, extrinsic self-healing cannot be sustained. While enclosed in the microcapsules, the healing agent exists in the uncured state. However, when released during the first healing, the healing agent is irreversibly cured, preventing repetitive self-healing.

To overcome these problems, later researchers developed intrinsic self-healing methods. These methods can be categorized into two types. Associative self-healing methods include hydrogen bonding, host–guest interactions, or exchange reactions involving transesterification, disulfide metathesis, transalkylation, transcarbamoylation, or azomethine metathesis [9,10,11,12,13]. Dissociative self-healing methods are based on the Diels–Alder reaction and a reversible urethane or hindered urea group [14,15,16,17,18]. Especially, Cao et al. studied thermal self-healing of thermosetting PU based on the reversible features of phenolic urethanes [18]. However, the mechanical and self-healing properties usually exhibit an inverse relationship. Polymers with aromatic disulfide groups can self-heal even at room temperatures, but their mechanical properties are low. In this aspect, hydrogels and supramolecular polymers have been extensively studied [18,19,20,21]. However, polymers that self-heal via reversible urethane groups must be heated to above the dissociation temperature of reversible urethane groups (i.e., to above 200 °C). Hindering the urea groups may lower the dissociation temperature of the polymer by the bulk structure around the urea group, enabling self-healing at room temperature. Unfortunately, the mechanical properties of polymers that self-heal via the reversible urethane or urea groups are relatively poor [17,18]. Light-triggered self-healing polymers are also studied. In this case, inorganic fillers such as carbon nanotubes (CNTs), gold nanoparticles (AuNPs), and graphene, heating is assisted by light. Polymers heated by light can undergo self-healing via the dissociation of hydrogen bonds or melting of crystals [22,23,24]. Polymer self-healing is also affected by various factors that influence the diffusion properties of polymers, such as hydrogen bonding, phase mixing, and crystallization. The hardness of polymers that self-heal by disulfide metathesis is very low [21]. Recently, polymers showing high self-healing efficiency even at low temperatures have been reported. Scratched PU film comprising poly(tetramethylene ether glycol), isophorone diisocyanate, and aromatic disulfide was completely healed after only 30 min at 40 °C [25]. Yanagisawa and co-workers reported complete healing of urethane elastomer at 36 °C after 24 h via hydrogen bonding of thiourea [26]. Leibler and co-workers reported that the self-healing properties of polymers were directly relatable to their stress relaxation times [27,28,29]. 

However, applicability of the above-mentioned healable polymers with reversible urethane bonds are limited by various conditions. For instance, polymers that self-heal only at high temperatures under inert gas conditions are unsuitable for low-temperature applications. In the present study, we investigate whether the concentration of the dynamic covalent bonds influences the self-healing property of PU. Typical urethane bonds constitute aliphatic hydroxyl and isocyanate groups, which are stable and indecomposable below 200 °C. The phenolic hydroxyl group can also react with the isocyanate group, blocking the isocyanate and improving the storage stability of the prepolymer. Urethane groups constructed from phenolic hydroxyl and isocyanate can re-dissociate into phenol and isocyanate via a reversible reaction. 

Inspired by these facts, we chose vanillyl alcohol (VA) as the diol and mixed it with a polyol to synthesize a VA-based PU with two hydroxyl groups (a phenolic hydroxyl group and an aliphatic hydroxyl group) that self-heals under heating. VA enables the introduction of a reversible urethane group composed of a phenolic hydroxyl with an isocyanate group into PU. The mechanical and self-healing properties of PUs with different concentrations of the dynamic chemical bonds are investigated. In addition, the self-healing properties of PU are improved by a modified chain extender (m-CE) that improves the mobility of the molecules.

## 2. Results and Discussion

### 2.1. Characterization of Vanillyl Alcohol (VA)-Based Polyurethane Elastomers (PUs)

Figure 1 shows the Fourier-transform infrared (FT-IR) spectra of the raw materials of the VA-based PUs. Hydroxyl groups and isocyanates were confirmed at 3000–3500 and 2270 cm^−1^, respectively. Hydrogen-bonded and free hydroxyl groups of VA were observed at 3340 and 3160 cm^−1^, respectively. VA-based PUs synthesized by the pre-polymer method are illustrated in Scheme 1. The peak at 2270 cm^−1^ in the FT-IR spectrum of the VA-based PUs disappeared after curing at 110 °C for 24 h (see Figure 2). Figure 2(A) shows the FT-IR spectra of representative PUs prepared from VA and m-CE, and Figure 2(B) magnifies these spectra in the 2000–1500 cm^−1^ wavenumber region. The absorption peaks of the hydrogen-bonded and free carbonyl groups in the PUs appeared at 1703 and 1733 cm^−1^, respectively. The magnitude of the hydrogen-bonded carbonyl peak increased with increasing VA content (see Appendix A). The I_1703_/I_1733_ ratios were 1.33, 1.77, 1.85, 1.89, and 2.21 for the control PU, VA10, VA20, VA30, and VA40, respectively, implying that increasing the VA content increased the hydrogen bonding in the PUs. However, the hydrogen-bonded carbonyl peaks in the VA40-5 and VA40-10 PUs were largely suppressed by the m-CE. Specifically, the I_1703_/I_1733_ ratios were 0.95 and 0.92 for VA40-5 and VA40-10, respectively. This result implied that hydrogen bonding was decreased, so the molecules became more mobile. Table 1 summarizes the average molecular weights of the synthesized PUs, determined by gel permeation chromatography (GPC). The decrease in average molecular weights with increasing VA content was attributable to the low reactivity of the phenolic hydroxyl groups of VA. On the other hand, the m-CE increased the average molecular weight of the VA-based PUs, owing to the catalytic effect of the tertiary amine of m-CE. 

Cao et al. [18] confirmed the reversibility of urethane groups formed by the phenolic hydroxyl group and isocyanate by FT-IR spectroscopy. To confirm the regenerated isocyanate groups, a model compound was synthesized by reacting two moles of 4,4’-methylene diphenyl diisocyanate (MDI) with three moles of VA. The complete consumption of isocyanate groups in the model structure was confirmed by titration following ASTM D2572-91. Additionally, the hydroxyl values of the model structure were 116.3 mg KOH/g. The hydroxyl value was determined according to ASTM 4274D. FT-IR spectra of the model compound could be obtained at elevated temperatures by heating the sample block and are given in Appendix A. In order to prevent the reaction of NCO groups and moisture in the air, FT-IR spectra were collected from the samples in an N_2_ gas environment inside the sample block. The characteristic peak of isocyanate groups generated by the reversible properties of urethane groups appeared at 2270 cm^−1^ above 140 °C in FT-IR spectra.

### 2.2. Microdomain Structure of VA-Based PUs

VA-based PUs synthesized in this study appeared opaque, mainly because of the crystallites formed by the hard segments. The microphase-separated structures of the VA-based PUs strongly depended on the VA/PTMEG1000 ratio. Figure 3 shows the small-angle X-ray scattering (SAXS) profiles of the control PU and VA-based PUs. The SAXS profiles confirmed the scatterings of hard and soft segments domains. The interdomain distance of hard domain in the SAXS profiles was ~20 nm, similar with the interdomain distance of the hard domains in previous studies of PUs [30,31,32,33]. The peak positions of the VA-based PUs appeared in a higher q range than in the control PU, indicating a lower interdomain distance of the VA-based PUs than that of the control PU. The interdomain distances were 19, 18, 18, and 17 nm, respectively, in the VA10, VA20, VA30, and VA40 samples; and they were 15 and 14 nm, respectively, in the VA40-5 and VA40-10 samples (in which phase mixing was induced by the m-CE). VA-based PUs exhibited much broader widths than the control PU (Figure 3 and Appendix A), implying that micro-phase separation and crystallinity were hindered by VA as well as by m-CE. 

### 2.3. Thermal Analyses of PUs Based on VA

The thermal stabilities of the VA-based PUs were analyzed by a thermogravimetric analyzer (TGA) (Figure 4 and Appendix A). VA-based PUs were less thermally stable than the control PU. More specifically, the 5% decomposition temperature of the Control PU was 311 °C, whereas those of the VA-based PUs decreased from 303 °C to 298 °C as the VA content increased from 10% to 40%. That is, the thermal stability decreased with increasing VA content. According to Zoran et. al., increasing the hard segment decreased the thermal decomposition temperature of segmented PUs as the hard segment was less thermally stable than the soft segment [34]. Increasing the VA content increased the proportion of hard segment in the VA-based PUs, thereby reducing their thermal stabilities in comparison with the control PU. However, in the VA-based PUs incorporating m-CE, the thermal stability was hardly changed by increasing the VA concentration. Weight loss due to volatile small molecules released by thermal degradation was not observed below 200 °C. The TGA results are listed in Appendix A.

Dynamic mechanical properties in the PUs were investigated by a dynamic mechanical analyzer (DMA). The results are displayed in Figure 5 and Appendix A. The soft-segment domains of the PUs underwent glass–rubber transitions below room temperature. The glass transition temperature of the soft-segment domains (*T*_gs_) manifested as a peak in the tan delta curve. In Appendix A, the *T*_gs_ increased with increasing VA concentration in the PU, reflecting the decreased content of the polyol constituting the soft domain and the increased content of the urethane group capable of hydrogen bonding (per unit mass). Meanwhile, the *T*_gs_ of the PUs with m-CE in Figure 5 decreased because the hydrogen bonding and hard segment packing were disturbed by the m-CE side-chain. In Appendix A, the flow temperature (*T_flow_*), at which the storage modulus of the PUs decreased after the rubbery plateau region, was lower in the VA-based PUs than in the control PU, and it increased with increasing VA content. This implied that the VA hindered the microphase separation, and dissociation of reversible urethane groups was lower than T_flow_ of the control PU. However, T_flow_ in the VA-based PU increased as the VA content increased because the hard segment concentration increased. 

At low temperatures, hydrogen bonds increased the storage modulus of the VA-based PUs over the control PU (as confirmed by FT-IR). In Figure 5, the m-CE also decreased the storage modulus (which was below *T*_gs_) and *T*_flow_ of the VA-based PUs, again by interfering with the hydrogen bonding and hard segment packing (confirmed by FT-IR). The stress relaxations in the PUs were investigated by employing DMA, and the test results are given in Figure 6 and Appendix A. Appendix A summarizes the relaxation times at which the storage moduli reached 1/e of their initial values. At 140 °C the relaxation time was 539 s in the control PU and 71, 102, 120, and 162 s in the VA10, VA20, VA30, and VA40 samples, respectively. The relaxation times of the VA-based PUs were lower than in the control PU, but they increased with increasing VA content. This trend was attributable to the decreasing content of PTMEG1000 (which constituted the soft segment) and increased disturbance of the microphase separation as the VA content increased. Self-healing of polymers has been frequently investigated in stress-relaxation tests, and stress-relaxation time was closely related with self-healing properties [20,21,25,26]. Figure 7 shows Arrhenius plots of the relaxation times of the VA-based PUs [29,34]. The relaxation times of the VA-based PUs indeed fitted the Arrhenius law, and the activation energies of the VA-based PUs (determined from the Arrhenius slopes) were lower than that of the control PU, but they increased with increasing VA content because molecular mobility of VA-based PUs decreased as a result of the increasing hydrogen bond and hard segment. However, the activation energy of VA40-10 (with 10% m-CE) reached 108.5 kJ/mol. The activation energy of VA40-10 decreased to 63.4% compared with VA40 because mobility of molecules was improved by the introduction of m-CE. 

Next, VA-based PUs were investigated by differential scanning calorimetry (DSC). *T*_gs_ increased with increasing VA content. More specifically, *T*_gs_ was −45 °C in the control PU and −43, −40, −36, and −20 °C in the VA10, VA20, VA30, and VA40 samples, respectively. The hard domain melted at approximately 150 °C. However, when the hydrogen bonds were disrupted by m-CE, *T**_flow_* decreased. For instance, the *T*_gs_ and melting temperature of VA40-10 were −30 °C and 143 °C, respectively. The increase in *T*_gs_ with VA content was attributable to phase mixing and hydrogen bonding by VA (the DSC results are displayed in Appendix A and summarized in Appendix A.) 

### 2.4. Mechanical Properties of the VA-based PUs 

Stress–strain properties of the PUs were studied in tensile tests, and they are presented in Figure 8 and Appendix A. Interestingly, increasing the VA content improved the tensile strengths and moduli of the PUs, but it lowered the elongation at break. Figure 8 shows representative tensile properties of the synthesized PUs. The molecular weight of the VA-based PUs decreased with increasing VA content. At first glance, this trend seemed incompatible with the increasing tensile strength of VA-based PUs with higher VA content, but it can be explained as follows. When the VA content increased, the decreased molecular weight and microphase separation in the VA-based PUs were more than offset by strong hydrogen bonding; consequently, the tensile strength and modulus improved.

### 2.5. Self-Healing Properties of the VA-based PUs 

Although VA-based PUs showed earlier stress relaxation at elevated temperatures (above 120 °C) than at low temperatures, their tensile properties at room temperature were superior to those of the control PU. Self-healing properties of the samples were evaluated on dog-bone specimens. The center of each prepared specimen was cut with a knife, and the two halves were immediately reattached. The self-healing process is imaged in Appendix A. The specimen was healed in a convection oven at 140 °C for 30 min. Tensile properties of the VA-based specimens at elevated temperatures after cutting are summarized in Appendix A. The healing efficiency decreased with increasing concentration of VA in the PU. Figure 9 shows the tensile properties of representative PUs before and after healing. VA-based PUs healed less effectively than the control PU because of the strong hydrogen bonds and close packing of hard segment, which lowered the mobility of the molecules. More specifically, self-healing efficiency was 69.3% in the control PU and 60.5%, 50.7%, 42.9%, and 32.4% in the VA10, VA20, VA30, and VA40 samples, respectively. However, incorporating the m-CE dramatically improved the healing efficiency of samples with the same VA concentration. Owing to the higher molecular mobility of molecules, the healing efficiency reached 88.4% and 96.5% in VA40-5 and VA40-10. The side chain of m-CE hindered hydrogen bonding among the urethane groups. Results of repetitive self-healing tests are displayed in Appendix A. After the second healing cycle, the healing efficiency of the control PU was 43.3% (decreased from 69.3% after the first cycle), but that of VA40-10 was 87.6% (down from 96.5% after the first cycle). This result confirmed that the m-CE dramatically improved the healing efficiencies of VA40-10. 

## 3. Materials and Methods

### 3.1. Materials

Butyl glycidyl ether (BGE, Mw: 130.19, 95%), diethanolamine (DEA, Mw: 105.14, 99%), poly(tetramethylene ether)glycol (Mw: 1,000, 99%) (PTMEG1000), and 1,4-butanediol (BDO; 99%) were obtained from Sigma–Aldrich. Vanillyl alcohol (VA, Mw: 154.17, 98%) was purchased from Tokyo Chemical Industry in Korea. 4,4’-methylene diphenyl diisocyanate (MDI, 99%) was obtained from BASF (BASF Korea, Yeosu, Korea).

### 3.2. Preparation of Modified Chain Extender

The m-CE was obtained by reacting DEA (10.5 g, 0.1 mol) with BGE (13 g, 0.1 mol). The mixture was stirred at room temperature for 24 h. The mixture was initially opaque but gradually became transparent through the exothermic reaction. Appendix A shows the FT-IR spectra of DEA, BGE, and m-CE. The epoxy peak at 915 cm^−1^ completely disappeared after the reaction. In the H^1^-NMR spectrum (Appendix A), the peaks at 3.6 and 3.8 ppm were attributable to the primary and secondary hydroxyl groups, respectively. Complete consumption of the secondary amine was confirmed by amine titration following ASTM D2074.

### 3.3. Preparation of Control PU and VA-based PUs

VA-based PUs were synthesized by the prepolymer method. The prepolymer was synthesized by reacting 0.1 mol of PTMEG1000 and 0.2 mol of MDI at 70 °C for 3 h. The synthesized prepolymer was mixed with the desired amounts of MDI, BDO, m-CE, and VA. After degassing, the mixtures were poured into a glass mold to prepare PU films and cured at 110 °C for 24 h in a convection oven. PU films obtained after the cure were employed for characterizations.

Table 2 lists the compositions of the VA-based PUs. The middle and final digits of the sample codes VA40-5 and VA40-10 denoted the molar percentages of VAs in the polyols and m-CE in the BDO, respectively.

### 3.4. Characterization

Synthesis of the m-CH and VA-based PUs were confirmed by Fourier-transform infrared (FT-IR) spectroscopy (FT-IR-302; Jasco,). H^1^-NMR spectra were recorded in chloroform-*d* with a Bruker AM400 spectrometer (600 MHz). Molecular weights of the VA-based PUs were determined by GPC (Agilent 1200S; Agilent) employing a refractive index detector (Optilab rEX; Wyatt). Samples were dissolved in DMF/THF (1:1 wt./wt.), and polystyrene standards were used for universal calibration. The flow rate of the GPC measurement was 1.0 mL/min. Thermal properties of the VA-based PUs were measured by DSC (Q20; TA Instruments). The samples were initially cooled to −90 °C and maintained at this temperature for 1 min. In all measurements, the temperature was ramped up to 240 °C at 10 °C/min. A second scan was conducted under the same conditions. All tests were performed in an N_2_ atmosphere. Dynamic mechanical properties and relaxation times of the VA-based PUs were measured by DMA (Q800; TA Instruments) using a film tension clamp. The films for DMA were prepared by carefully cutting the cured PU films into 5.3 mm-wide sections. Dynamic mechanical properties were measured between −90 and 240 °C inclusive at a heating rate of 5 °C/min. Relaxation tests were also performed at 110, 120, 130, 140, 145, and 150 °C for 20 min under an axial force of 0.01 N and a deformation of 1%. Tensile strengths and percentage elongations were measured by a universal testing machine (LR5K UTM; Lloyd) following the ASTM-D638 method. SAXS patterns (D8 DISCOVER, Bruker) were collected at room temperature using Cu–Kα radiation (*λ* = 1.541 Å) at a voltage of 50 kV and a current of 1000 µA. The Bragg angle (2*θ*) ranged from 0° to 9°. 

## 4. Conclusions

This study investigated the self-healing of VA-based PUs via reversible urethane bonds formed by the reaction of phenolic hydroxyl and isocyanate groups, which can be efficiently exploited for self-healing PUs. In particular, we investigated the influence of VA and m-CE contents on the self-healing properties by measuring the thermal and dynamic mechanical properties of the prepared samples. We confirmed that the urethane bonds can be reversibly dissociated and that the isocyanate groups regenerated above 140 °C. As the VA content of the polyols increased from 10% to 40%, the relaxation time reduced from that of the control PU. Increasing the VA content decreases the self-healing efficiency by increasing the hydrogen bonding of hard segments and decreasing the proportion of soft segments, thereby reducing the mobility of the molecules. To overcome this problem, molecular mobility was increased by incorporating m-CE into the VA-based PUs. Hydrogen bonding and hard domain packing, which improved the mechanical properties but hindered the molecular movements, was reduced by the m-CE. Consequently, the I_1703_/I_1733_ ratio in the FT-IR spectrum, which indicated the extent of hydrogen bonding in the PU, significantly decreased to 0.92 in VA40-10. On the other hand, the I_1703_/I_1733_ value of VA40 (without the m-CE) was 2.21, indicating strong hydrogen bonds in this sample. The interdomain distance in the SAXS profile of VA40-10 was only 14 nm, reflecting decreased microphase separation. Meanwhile, the relaxation time of the VA-based PUs at 140 °C increased from 71 to 162 s with increasing VA content. But after incorporating the m-CE, it significantly decreased, reaching 65 and 59 s in the VA40-5 and VA40-10 samples, respectively. This result was also closely related to microphase separation and hydrogen bonding in the PUs. Thus, disrupting the hydrogen bonds and microphase separation notably affected the self-healing properties of the PUs. Reduced hydrogen bonds and lower microphase separation improved the mobility of the molecules. Thus, reducing the molecular weight by dissociating the reversible urethane bonds efficiently contributes to the self-healing of PUs. To ensure efficient PU self-healing, the molecular structure design should increase both the molecular mobility and the concentration of dynamic covalent bonds.

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
