# Peer review of "Introduction of Reversible Urethane Bonds Based on Vanillyl Alcohol for Efficient Self-Healing of Polyurethane Elastomers"

_molecules, 2019, doi:10.3390/molecules24122201_

Round 1
Reviewer 1 Report
The authors prepared self-healing polyurethane vanillyl alcohol. Some information and explanation of this study are still lacked. In my opinion, the authors should major revision of the manuscript in order to “Molecules” because of the following comments.
1. Please define “self-healing” polyurethane and compare to other “self-healing” materials.
2. Line 12 “The urethane groups formed by reacting phenolic hydroxyl groups with isocyanates are known to be reversible at high temperatures.” Please find the reference to support the phenomenon.
3. Scheme 1. Please check the structure of PTMEG. Do “COOH” be the functional group of PTMEG?
4. How to check the structure of from heating to cooling? NCO groups of MDI and prepolymer are easy to react with water.
5. The authors didn’t mention the details of the method. Especially, prepared method of films is required.
6. The TGA results showed the thermal stability of controlled polyurethane higher than vanillyl alcohol-based polyurethane. Please explain it and add the differential thermogravimetric analysis.
7. There are many typographical and grammatical errors in the manuscript.
Author Response
Invaluable comments to improve the manuscript are appreciated. Our responses to the comments are described in the attached file.

Reviewer 2 Report
In this manuscript, the authors reported a class of self-healing VA-based polyurethanes via introducing reversible urethane bonds composed of phenolic hydroxyl and isocyanate groups. Moreover, the properties of materials and VA concentration were further investigated. The paper is well-written and the results are very interesting. However, some descriptions of results and experiment are not clear. Therefore, I would like to reconsidered this paper to be published after major revisions.
Suggestions on how to improve this paper are listed below:
The introduction and background information are not adequate. There’re many examples reported self-healing PU and composited materials via photo/light treatment. Please add and check the related references such as: J. Mater. Chem. A, 2016, 4, 10683; J. Mater. Chem. C, 2016, 4, 5932; Chem. Soc. Rev., 2013, 42, 7244.
Can you explain why the PDI of VA10 is much larger than other polymers?
It is necessary to provide the tan delta curve of DMA results and listed the Tg for each polymers.
The description of Figure 9 is not clear. The description of number percent should be provided.
The ‘140oC’ in Figure 9 will lead confused. Is that mean the tensile test were measured under 140 degrees?
Author Response
Invaluable comments to improve the manuscript are highly appreciated. Our responses to the comments are described in the attached file.

Round 2
Reviewer 1 Report
My comments have been modified accordingly. This article could be accepted in present form.
Reviewer 2 Report
The revised paper well addressed all kinds of suggestion and question. Therefore, I would like to recommend this paper to be published in present version.